# Biotechnological Potential of Microorganisms for Mosquito Population Control and Reduction in Vector Competence

**DOI:** 10.3390/insects14090718

**Published:** 2023-08-22

**Authors:** Ricardo de Melo Katak, Amanda Montezano Cintra, Bianca Correa Burini, Osvaldo Marinotti, Jayme A. Souza-Neto, Elerson Matos Rocha

**Affiliations:** 1Malaria and Dengue Laboratory, Instituto Nacional de Pesquisas da Amazônia-INPA, Manaus 69060-001, AM, Brazil; ricardokatak@gmail.com; 2Multiuser Central Laboratory, Department of Bioprocesses and Biotechnology, School of Agricultural Sciences, São Paulo State University (UNESP), Botucatu 18610-034, SP, Brazil; amanda.cintra@unesp.br (A.M.C.); jsouzaneto@vet.k-state.edu (J.A.S.-N.); 3Florida Medical Entomology Laboratory, University of Florida, Vero Beach, FL 32962, USA; bianca.kojin@ufl.edu; 4Department of Biology, Indiana University, Bloomington, IN 47405, USA; omarinotti@gmail.com

**Keywords:** biotechnology, microorganisms, bacteria, fungi, vector control, mosquitoes

## Abstract

**Simple Summary:**

Mosquitoes carry pathogens that can cause diseases like malaria, dengue fever, chikungunya, yellow fever, and Zika fever, causing more than 700,000 deaths each year around the world. Chemical insecticides kill mosquitoes effectively, minimizing the spread of illnesses. However, these chemicals have disadvantages such as high production costs and negative impacts on the environment and other organisms, including humans. Furthermore, mosquitoes are becoming more resistant to chemical insecticides. Therefore, alternatives to commonly used insecticides are urgently required. In this review, we highlight the biotechnological potential of microorganisms to control vector mosquitoes and reduce disease transmission. In addition, we emphasize the importance of more basic research and improved translational research methods to bridge the gap between academic research on bioinsecticides and public health interventions.

**Abstract:**

Mosquitoes transmit pathogens that cause human diseases such as malaria, dengue fever, chikungunya, yellow fever, Zika fever, and filariasis. Biotechnological approaches using microorganisms have a significant potential to control mosquito populations and reduce their vector competence, making them alternatives to synthetic insecticides. Ongoing research has identified many microorganisms that can be used effectively to control mosquito populations and disease transmission. However, the successful implementation of these newly proposed approaches requires a thorough understanding of the multipronged microorganism–mosquito–pathogen–environment interactions. Although much has been achieved in discovering new entomopathogenic microorganisms, antipathogen compounds, and their mechanisms of action, only a few have been turned into viable products for mosquito control. There is a discrepancy between the number of microorganisms with the potential for the development of new insecticides and/or antipathogen products and the actual available products, highlighting the need for investments in the intersection of basic research and biotechnology.

## 1. Introduction

Microorganisms constitute a large group of genetically diverse biological entities found in a wide range of terrestrial and aquatic habitats, playing crucial roles in the balance of ecosystems [1,2,3]. Advances in microbiology, molecular biology, and genomics enabled the biotechnological exploration of microbes, allowing the discovery and production of antibiotics [4,5], foods [6,7], alcoholic beverages [8], bioremediators [9,10], fertilizers [11], and biopesticides [12,13]. The microorganisms associated with mosquitoes have drawn special attention to their potential applications in public health (Figure 1) [14,15,16]. In this review, we highlight the largely unexplored potential of microbes for the control of mosquito-borne diseases and the need for improved translational research strategies, encouraging efforts toward bridging the gap between academic research and public health interventions.

## 2. Bacteria for Biological Control of Medically Important Mosquitoes

Today, chemical insecticides are used as the main tool for mosquito control [17,18] but are no longer as effective as in the past due to the selection of insecticide-resistant individuals in mosquito populations worldwide [19,20,21,22]. Furthermore, chemical insecticides harm the environment, contaminating groundwater systems through infiltration into the soil, reaching riverbeds, accumulating in fish and other animals [23], and through spraying, contaminating the air, affecting human health [24,25]. These facts emphasize the prominent need to develop new, efficient, and environmentally safe tools for the control of vector mosquitoes and the diseases they transmit.

Bacteria of the Bacillaceae family infect insects and produce toxins with insecticidal properties. *Bacillus thuringiensis israelensis* (Bti) and *Lysinibacillus sphaericus* (Lbs), the latter formerly known as *Bacillus sphaericus*, are widely known for their larvicidal activity against several species of mosquitoes [26,27,28,29,30]. Due to their high efficacy, safety, and the well-characterized mechanisms of action of their toxins, several strains of Bti and Lbs are included in commercially available biological larvicide formulations endorsed by organizations such as the World Health Organization [31] and the Environmental Protection Agency (EPA) in the United States of America (www.epa.gov/mosquitocontrol/bti-mosquito-control) (accessed on 12 April 2023).

The toxins of Bti and Lbs and the mechanisms associated with mosquito mortality have been extensively studied [32,33]. Briefly, in Bti, the molecules responsible for the entomopathogenic action are mainly the crystal toxins Cry4Aa, Cry4Ba, Cry10Aa, and Cry11Aa, the cytolytic toxins Cyt1Aa, and Cyt2Ba, and the P19 and P20 proteins [34,35]. Cry and Cyt toxins, also known as δ-endotoxins, when present in the midgut of mosquito larvae, are proteolytically activated by digestive proteases, bind to specific receptors on the host cell membranes, and cause cell rupture resulting in the death of infected larvae [36]. The Lbs Bin toxins [37,38,39], Mtx [40], Cry48Aa, and Cry49Aa [41] display entomopathogenic mechanisms similar to those described above for the Bti toxins [42]. Acting synergistically, these toxins result in effective and potent toxic activity against mosquitoes [43,44,45].

The metabolites of the *Saccharopolyspora spinosa* bacterium are also useful in the control of mosquitoes. Spinosad is a mixture of two *S. spinosa* metabolites, spinosyn A and spnosyn D, endorsed by the US EPA in 1997 [46,47,48]. This product operates uniquely by affecting post-synaptic nicotinic receptors related to acetylcholine and gamma-aminobutyric acid. This stimulation leads to involuntary muscle contractions, prostration with tremors, and paralysis in insects [49]. With minimal environmental and human health risks, Spinosad is widely used in integrated pest management, countering worldwide insecticide resistance [48].

Finding new microbes with larvicidal activities similar to those of Bt, Lbs and *S. spinosa* has been the goal of several research groups around the world. However, this endeavor has been limited by the fact that culture media do not always meet the requirements for the growth of many species of bacteria [50]. Therefore, the search for entomopathogenic bacteria is often limited to those that grow in commercially available culture media. Despite these limitations, bacterial strains that are suitable for cultivation and have larvicidal activity have been identified (Table 1).

Unfortunately, most of them have not been further investigated, developed for applicable products, or tested under field or semi-field conditions. Additional research to understand their mechanisms of action, effects on non-target organisms, and potential for large-scale production is needed. Toward these goals, live bacteria, deactivated bacteria, and fractionated cells or culture media have been tested for their larvicidal activities. For example, *Bacillus safensis*, *Bacillus paranthracis*, and *Bacillus velezensis* culture supernatants and crude lipopeptide extracts were shown to be toxic to *Aedes aegypti* (Linnaeus, 1762) [51]. Whole genome sequencing and mass spectrometry analysis of those isolated bacteria strains revealed that these microorganisms synthesize bacteriocin, beta-lactone, and terpenes potentially toxic to mosquito larvae [51]. Nineteen *Bacillus* sp. strains and two strains of *Brevibacillus halotolerans* isolated from Amazonian environments showed larvicidal activity against *Ae. aegypti* [52]. The supernatant and pellet fractions of those strains were tested separately, revealing that cellular and secreted metabolites are toxic to mosquito larvae; however, the active molecules were not identified. In another study, *Bacillus* sp. EG6.4 culture exhibited high toxicity to *Ae. aegypti* larvae. Transmission electron microscopy revealed subterminal oval-shaped endospores and massive parasporal inclusions unrelated to cry toxin. The bacterium showed hemolytic activity, indicating its potential to produce biosurfactants. Detection of the surfactin-coding gene (srfA-D) confirmed its association with *B. mojavensis* strain PS17. These findings suggest that biosurfactant production, particularly surfactin, may play a role in *Bacillus* sp. EG6.4’s larvicidal mechanism [53]. The testing of whole or fractionated bacteria and culture media is useful for defining procedures and formulation of new biolarvicides.

**Table 1 insects-14-00718-t001:** Bacterial strains active against *Aedes*, *Culex*, and/or *Anopheles* mosquito larvae.

Bacterium	Toxic Formulation	Target Mosquito Genera			Refs.
		*Aedes*	*Culex*	*Anopheles*	
*Bacillus thuringiensis* var. *israelensis* (Bti)	Extract (spores and crystals)	+	−	−	[54]
	Sporulated culture powder (tablet formulation XL-47)	+	−	−	[55]
	Spores and crystals tablet	+	−	−	[56]
	Spores and crystals tablet	+	−	−	[57]
	VectoBac WG *	+	−	−	[58]
	Formulated product	−	+	−	[59]
	Binary mixtures (Bti plus Deltamethrin)	−	+	−	[60]
	Cry2Aa and Cyt1Aa crystals	−	+	−	[61]
	Crystallogenic variants.	+	+	−	[62]
	Two recombinant proteins (Cry10Aa and Cyt2Ba)	+	−	−	[63]
	Xpp81Aa toxin combined with Cry2Aa and Cry4Aa	+	−	−	[35]
	Kappa-carrageenan and Vectobac 12 AS hydrogels *	+	−	−	[64]
	Bti extracts	+	−	−	[65]
	Vectobac^®^ AS * + and plant-ethanol extracts	−	+	−	[66]
	Granular formulation (Vectobac G) *	−	−	+	[67]
	Dispersible granule (strain AM65-52) *	+	+	+	[68]
	Bti strain Becker Microbial products (BMP) *	−	+	+	[69]
	Bti sprayed products BACTIMOS WP^®^ *; VECTOBAC TP^®^ *; and TEKNAR^®^ HP-D *	−	−	+	[70]
	Bti water dispersible granular (WDG) VectoBac@ strain AM65-52 *	−	−	+	[71,72,73,74]
*Bacillus thuringiensis* (Other strains)	Total and lyophilized culture	+	+	−	[75]
	Bacterial cultures	+	+	+	[76]
	Bacterial suspensions (spores and crystals)	+	−	−	[77]
	Spores	+	+	+	[78,79]
	Parasporal crystalline inclusion bodies	+	−	−	[35]
	Culture supernatant	+	−	+	[29]
	Synergistic interaction (Purified Cry11Aa and Cyt1Aa Toxins)	+	−	−	[80]
	Synergistic action of the Cry and Cyt proteins	−	−	+	[81]
*Lysinibacillus sphaericus* (Lbs)	Culture supernatant	+	−	+	[29]
	Spores and vegetative cells	+	+	−	[30]
	Cell suspension plus glyphosate	+	−	−	[82]
	Spore crystals (lyophilized powder)	+	+	+	[83]
	Spores	-	+	-	[84]
	Granular formulation (Vectobac G) *	−	+	+	[67]
	Vectolex G *	−	+	−	[85]
	S-layer protein	−	+	−	[86]
	Purified BinA and BinB proteins	−	+	+	[87,88]
	Spore crystals and purified S-layers protein	−	+	+	[89]
	Synergy of Mtx and Cry proteins	−	+	−	[44]
	Purified BinA and BinB proteins	−	+	−	[39]
	VectoLex^®^ * WG plus Pyrethroid Resigen^®^	−	+	−	[90]
	Cry48Aa and Cry49Aa proteins combined	−	+	−	[91]
	Synergistic interaction (S-Layer and spores/crystals)	−	+	−	[92]
	VectoLex (ABG-6185) *	−	−	+	[93]
	Suspension (lyophilized bacteria)	−	−	+	[94]
	VectoLex^®^ CG *	−	−	+	[95]
	Bin toxin proteins	−	−	+	[96]
*Acidovorax* sp.	Cell-free supernatant	+	−	−	[97]
*Aneurinibacillus aneurinilyticus*	Bacterial suspension	+	+	+	[98]
*Bacillus amyloliquefaciens*	Biosurfactant	+	+	+	[99]
*Bacillus cereus*	Culture supernatant	+	−	+	[29]
*Bacillus circulans*	Spores	+	+	+	[100]
*Brevibacillus halotolerans*	Supernatant and pellet fractions of bacterial cultures	+	−	−	[52]
*Bacillus licheniformis*	Dahb1 exopolysaccharide (Bl-EPS)	+	−	+	[101]
*Brevibacillus laterosporus*	Suspension of sporulated cells	+	−	+	[102]
	Spore and the canoe-shaped parasporal body (CSPB) structure	+	−	−	[103]
	Purified protein crystals	+	−	+	[104]
	Pellets (cells and spores)	+	−	−	[105]
	Spores	+	−	−	[106]
*Bacillus paranthracis*	Pellets (cells)	+	−	−	[51]
*Bacillus safensis*	Supernatant and pellet fractions of bacterial cultures	+	−	−	[52]
	Pellets (cells)	+	−	−	[51]
*Bacillus subtilis*	Culture supernatant	+	−	+	[29]
	Crude cyclic lipopeptides (CLPs)	−	+	−	[107]
	Crude surfactin	−	−	+	[108]
	Bacterial biomass	+	−	−	[109]
	Biosurfactants	−	−	+	[110,111]
*Bacillus megaterium*	Bacterial culture	+	−	−	[52]
*Bacillus nealsonii*	Secondary metabolites	+	−	−	[112]
*Bacillus tequilensis*	Cyclic lipopeptide Biosurfactant	−	−	+	[113]
*Bacillus velezensis*	Bacterial culture	+	−	−	[52]
	Pellets (cells)	+	−	−	[51]
*Chromobacterium* sp.	Hydrogen cyanide	+	−	+	[114,115]
*Chromobacterium anophelis*	Bacterial suspension	−	−	+	[116]
*Pantoea stewartii*	Silver nanoparticles	+	+	+	[117]
*Paraclostridium bifermentans*	Clostridial neurotoxin	−	−	+	[118]
*Peanibacillus macerans*	Bacterial biomass	+	−	−	[109]
*Photorhabdus luminescens*	Secondary metabolites (Culture fluids)	+	−	−	[119]
	Secondary metabolites	+	−	−	[120]
*Photorhabdus luminescens subsp. akhurstii*	Bacterial cell suspension	+	−	−	[121]
*Pseudomonas* sp.	Bacterial cell suspension	−	+	−	[88]
*Priestia aryabhattai*	Silver nanoparticles	+	+	+	[117]
*Serratia marcescens*	Prodigiosin	+	−	+	[122,123]
	Bacterial suspension	+	−	−	[124]
*Serratia nematodiphila*	Bacterial cultures	+	+	+	[76]
*Saccharopolyspora spinosa*	Spinosad (Tracer^®^) *	+	−	+	[125]
	Spinosad formulation	+	+	+	[46]
	Spinosad-based product (Laser^®^) *	+	+	+	[126]
	Spinosad SC (Tracer^®^) *	+	−	−	[127]
	Spinosad tablet (DT) and granules (GR) *	+	−	−	[128]
	Spinosad powder *	−	+	−	[129]
	Spinosad formulation *	−	−	+	[130,131]
	Natular T-30 formulation *	−	+	−	[132]
	Formulation emulsifiable Concentrate	−	+	−	[133]
*Streptomyces* sp.	Secondary metabolites	+	−	−	[112,134]
*Xenorhabdus indica*	Bacterial cell suspension	+	−	−	[121]
*Xenorhabdus nematophila*	Secondary metabolites	+	−	−	[120]
	Secondary metabolites (culture fluids)	+	−	−	[119]
*Xenorhabdus stockiae*	Bacterial cell suspension	+	−	−	[121]

This list is not exhaustive but provides ideas for future research and product development opportunities. The asterisks (*) indicate products already registered and authorized for commercial use.”+” sign indicates the toxic action of bacteria or their metabolic products on the target mosquito. “−” sign indicates absence of action due to the study not targeting the mosquito in question.

## 3. Fungi and Promising Approaches for Controlling Vector Mosquitoes and *Plasmodium* spp.

Fungi and their metabolites are potentially useful for the control of medically important mosquitoes [135,136,137,138,139,140]. In fact, fungal strains have already been applied as complementary measures for the control of vector mosquitoes [141,142,143,144]. The identification of mosquito fungi with larvicidal activities (Table 2), highlights the potential for their application as tools in mosquito management.

*Beauveria bassiana* strains infect and kill a variety of insects, including mosquitoes. Application of *B. bassiana* spores on surfaces where mosquitoes rest [145], the impregnation of spores in traps [146], the association of the fungus with insecticides, such as the combination of *B. bassiana* and permethrin [147], and the spread of the fungus by females mating with pre-inoculated males [148] have been proposed as means of field applications of *B. bassiana* against mosquitoes. The attraction of *An. stephensi* to spores of *B. bassiana* present in dead and dying caterpillars infected with the fungus [149] has been proposed as a useful alternative to infect mosquitoes. Furthermore, experimental evolution has been applied successfully to increase the efficacy of *B. bassiana* to *Anopheles coluzzii* (Coetzee et al., 2013) [150].

Exposure to lethal and sublethal doses of *B. bassiana* spores decreases adult *Ae. aegypti* and *Aedes albopictus* (Skuse, 1894) host-seeking behavior and fecundity [135,151]. Infected mosquitoes, while still alive, spread the fungus through the vector population. *Beauveria bassiana* spores and extracts are also effective against mosquito larvae [152,153,154]. As a result of this evidence, several strains of *B. bassiana* are authorized for use as biological insecticides, against vector mosquitoes, by regulatory agencies such as the EPA [155] and ANVISA in Brazil [156].

*Metarhizium anisopliae* is another fungus with biotechnological potential for mosquito control [157]. Its entomopathogenic mechanism is similar to that of *B. bassiana*. After contact, the spores germinate, producing hyphae, which penetrate the insect exoskeleton and develop inside the host’s body [158,159].

*Metarhizium anisopliae* CN6S1W1 is effective against *Ae. albopictus* and *Cx. pipiens* [160]. The fungus also affects the behavior of *An. gambiae* mosquitoes by inhibiting blood feeding and reducing fecundity and oviposition [161]. Concurrent infections with both *M. anisopliae* and *B. bassiana* shorten the lifespan of *Ae. aegypti* [145,162]. The synergistic actions of *M. anisopliae* and *B. bassiana*, together with the imidacloprid immunosuppressant, showed greater larvicidal activity against *Cx. quinquefasciatus* than the respective entomopathogens alone [163]. Suspensions of *M. anisopliae*-VKKH3 conidia were highly toxic to *Ae. aegypti*, *An. stephensi*, and *Cx. quinquefasciatus* larvae. It was observed that the larvae were completely invaded by conidia, which formed mycelium and were able to multiply and intensely invade the larval tissues, with the highest virulence observed at a dose of 10^10^ conidia/mL [138]. Ethyl acetate extracts from *M. anisopliae* are active against mosquito larvae. The metabolites contained in those extracts have not been identified but they may represent a solution for mosquito larvae control, since *M. anisopliae* conidia are not effective in the aquatic environment [164].

In addition to *B. bassiana* and *M. anisopliae*, other fungi have been reported with high biotechnological potential for mosquito control. The killing activity of *Aspergillus nomius* spores toward adult *Ae. albopictus* was comparable to those of *B. bassiana* [165]. Crude and purified extracellular extracts of *Aspergillus niger* with larvicidal action against *An. stephensi*, *Cx. quinquefasciatus*, and *Ae. aegypti* were reported [166]. Di-N-Octyl phthalate, (1H-Benzoimidazole-2-Yl)-[4-(4-Methyl-Piperazin-1-Yl)-Phenyl]-Amine, and 6,8-Dimethyl-5-Oxo-2,3,5,8-Tetrahydroimidazo [1,2-A] Pyrimidine, secondary metabolites of *Aspergillus flavus* and *Aspergillus fumigatus* [167] and preg-4-en-3-one, 17. α-hydroxy-17. β-cyano-, trans-3-undecene-1,5-diyne, and pentane, 1,1,1,5-tetrachloro-, from *Aspergillus tamarii* have been suggested to be responsible for larvicidal activity [168]. The biosafety of products derived from *Aspergillus* spp., or the fungus itself, still needs to be investigated. Suspensions of *A. flavus* conidia exhibited considerable toxicity against non-target organisms present in aquatic environments of mosquito larvae [169].

Species of the genus *Isaria* also have entomopathogenic characteristics for mosquito control. *Isaria tenuipes* [170], *Isaria javanica* ARSEF 5874, and *Isaria cateniannulata* ARSEF 6241 strains showed high levels of pathogenicity toward *Ae*. *aegypti* [137]. Larvicidal activity against *Cx. quinquefasciatus* and *Ae. aegypti* were demonstrated with silver nanoparticles (AgNps) carrying secondary metabolites of *Isaria fumosorosea* (Ifr) [171]. Other fungal species of interest that may be useful for vector control include *Trichoderma asperellum* [172] and *Hyalodendriella* sp. [173] which produce metabolites toxic to mosquitoes.

**Table 2 insects-14-00718-t002:** Fungal strains active against *Aedes*, *Culex*, and/or *Anopheles* mosquito larvae.

Fungus	Toxic Formulation	Target Mosquito Genera			Refs.
		Aedes	Culex	Anopheles	
*Beauveria bassiana*	Mycotrol ESO *	+	−	−	[174]
	Fungal suspensions	+	−	−	[145]
	Surfaces treated with conidia	+	−	−	[148]
	Spores	+	−	−	[135]
	Oil-formulated spores	−	−	+	[149]
	Fungal suspensions	−	−	+	[152]
	Spores	−	−	+	[150]
	Fungal suspensions	+	+	−	[175]
*Metarhizium anisopliae*	Conidial suspension	−	+	−	[176]
	Fungal conidia	+	−	−	[177]
	Fungal suspensions	+	−	−	[145]
	Conidial suspension	+	+	−	[160]
	Oil formulation	−	+	+	[178]
	Secondary metabolites	+	+	+	[179]
*Aspergillus niger*	Crude metabolites	+	+	+	[166]
*Aspergillus flavus*	Secondary metabolites	+	+	+	[167]
	Suspensions of conidia	+	-	−	[169]
	Culture filtrates	−	+	−	[180]
*Aspergillus fumigatus*	Secondary metabolites	+	+	+	[167]
*Aspergillus parasiticus*	Culture filtrates	−	+	−	[180]
*Aspergillus tamarii*	Endophytic fungal extracts	+	+	−	[168]
*Aspergillus terreus*	Mycelia (ethyl acetate and methanol extracts)	+	+	+	[181]
	Emodin compound	+	+	+	[182]
*Aspergillus nomius*	Spores	+	−	−	[165]
*Beauveria tenella*	Blastospores suspensions	+	+	−	[183]
*Cladophialophora bantiana*	Secondary metabolites	+	+	−	[184]
*Chrysosporium lobatum*	Secondary metabolites	−	+	+	[185]
*Chrysosporium tropicum*	Secondary metabolites	+	+	+	[186]
*Fusarium moniliforme*	Isoquinoline type pigment	+	−	+	[187]
*Fusarium oxysporum*	Temephos + *F. oxysporum* extract	+	+	+	[188]
*Fusarium vasinfectum*	Culture filtrates	−	+	−	[180]
*Isaria javanica*	Conidial suspensions	+	−	−	[137]
*Isaria cateniannulata*	Conidial suspensions	+	−	−	[137]
*Isaria tenuipes*	Conidial suspensions	+	−	−	[170]
*Isaria fumosorosea*	Secondary metabolites	+	+	−	[171]
*Paecilomyces* sp.	Secondary metabolites	+	+	+	[134]
*Penicillium daleae*	Mycelium extract	+	+	−	[189]
*Penicillium falicum*	Culture filtrates	−	+	−	[180]
*Penicillium marneffei*	Spores	−	+	−	[190]
*Penicillium* sp.	Ethyl acetate extract	−	+	−	[191]
	Ethyl acetate extract	+	+	−	[192]
*Pestalotiopsis virgulata*	Ethyl acetate mycelia (EAM) extracts and liquid culture media (LCM)	+	−	+	[193]
*Podospora* sp.	Sterigmatocystin compound	−	−	+	[194]
*Pycnoporus sanguineus*	Ethyl acetate mycelia (EAM) extracts and liquid culture media (LCM)	+	−	+	[193]
*Trichoderma asperellum*	Methanolic extract	−	−	+	[172]
*Trichoderma harzianum*	Mycosynthesized silver nanoparticles (Ag NPs)	+	−	−	[195]
*Trichoderma viride*	Culture filtrates	−	+	−	[180]
*Hyalodendriella* sp.	EtOAc extract	+	−	−	[173]
*Verticilluim lecanii*	Spores	−	+	−	[190]
	Conidia	−	−	+	[196]

This list is not exhaustive but provides ideas for future research and product development opportunities. The asterisks (*) indicate products already registered and authorized for commercial use.”+” sign indicates the toxic action of bacteria or their metabolic products on the target mosquito. “−” sign indicates absence of action due to the study not targeting the mosquito in question.

### The Potential of Fungi as Anti-Plasmodium Agents for Malaria Control

Fungi with potential antiparasitic properties, particularly against protozoa of the genus *Plasmodium*, have been researched as a potential tool to combat malaria. Endophytic fungi isolated from different organs of *Annona muricata*, a medicinal plant commonly used in traditional Cameroonian medicine against malaria, completely inhibited the growth of *P. falciparum* in vitro. Of the 152 fungi tested, 17.7% showed activities against different strains of the parasite, with the strongest effects from fungi belonging to the genus *Fusarium*, *Thricoderma*, *Aspergillus*, *Penicillium*, and *Neocosmopora* [197]. Compounds such as oxylipin and alternarlactones from *Penicillium herquei* and *Alternaria alternata*, respectively, demonstrated in vitro antiplasmodial activity [198,199]. A killer toxin purified from *Wickerhamomyces anomalus*, a symbiotic yeast of insects, when supplemented in a mosquito diet interfered with the development of ookines in the *An. Stephensi* midgut [200]. *Aspergillus* also showed antiplasmodial activity when supplemented in the mosquito diet. This activity was shown to be related to the inhibition of the interaction between parasites and fibrinogen-related protein-1 (FREP1), an agonist of gametocytes and ookinetes [201]. These authors identified the fungal metabolite, orlandin, as a candidate reagent to inhibit *P. falciparum* infection in *An. gambiae*.

The topical application or spraying of *B. bassiana* on the mosquito cage mesh killed ~92% of *An. stephensi* on day 14 after exposure and reduced the number of *Plasmodium chabaudi* sporozoite-positive mosquitoes. Although no impact on the early stages of the parasite (gametocytes and oocyst stages) was noted, the combined effect of mosquito mortality and reduced sporozoite prevalence was estimated to result in the reduction in malaria transmission risk by a factor of about 80 [202]. However, other studies did not reproduce these results and the differences in the outcomes could be related to the differences in experimental conditions and in both mosquito and parasite strains. Further consideration of these fungi in antimalarial campaigns would require further in-depth research to identify potential factors contributing to the observed differences in the impacts of *B. bassiana* and *M. anisopliae* on the development of *Plasmodium* species in *Anopheles* mosquitoes [203,204].

## 4. The Role of Insect-Microbiota Associations in Vector Competence

Associations between mosquitoes and their microbiota have gained significant attention in scientific research due to their impact on vector competence [205,206,207,208,209,210]. In the following, we discuss ways these associations can influence vector competence. Understanding these interactions is essential for developing effective vector-borne disease control strategies and reducing their impact on public health.

### 4.1. Symbiotic Bacteria and Their Potential against Infectious Agents

The mosquito microbiota influences host development, nutrition, reproduction, and immune responses to invading organisms [211,212,213,214]. While the composition of the mosquito microbiota is largely defined by the environment in which they live [215,216,217], resident bacteria can modulate the development and replication of parasites and viruses within their vectors [218,219,220,221,222,223,224,225]. Although this modulation can enhance or reduce the survival and replication of pathogens within mosquitoes, those mosquito–microbiota interactions that negatively affect pathogens offer possibilities to control arthropod-borne diseases.

For example, the Gram-negative bacteria, *Escherichia coli* H243, *E. coli* HB101, *Pseudomonas aeruginosa*, and *Ewingella americana* inhibit the formation of *Plasmodium falciparum* oocysts, in *Anopheles stephensi* (Liston, 1901) [226]. *Enterobacter* sp. (*Esp*_Z) isolated from the intestine of *Anopheles gambiae* (Giles, 1902) inhibited the development of malaria parasites when reintroduced into this same vector species [227,228]. The formation of oocysts of *Plasmodium berghei* was affected by the presence of *Serratia marcescens*-HB3 in *An. stephensi* [229]. In *An. gambiae*, *Escherichia coli*, *S*. *marcescens*, and *Pseudomonas stutzeri reduced* the prevalence and intensity of *P. falciparum* infection [230]. The *Serratia* Y1 strain exerts inhibitory activity on *P. berghei* ookinetes by activation of the Toll immune pathway in *An. stephensi* [231]. *Serratia ureilytica* (Su_YN1) produces an antimalarial lipase (AmLip) that inhibits the formation of *P. falciparum* oocysts in *An. stephensi* and *An. gambiae* [232]. *Asaia* SF2.1 also inhibits *Plasmodium* development in anophelines [233].

Virus replication in their vectors is also regulated by the mosquito microbiota. Bacteria of the genera *Proteus*, *Paenibacillus*, and *Chromobacterium* inhibited the replication of dengue virus serotype 2 (DENV-2) when administered to mosquitoes [114,234]. Some of the mechanisms by which symbiotic bacteria can hamper pathogen development have been elucidated and can be exploited to inhibit the spread of infectious agents by mosquitoes (Figure 2).

### 4.2. Wolbachia-Based Strategy for Controlling Mosquito-Borne Viruses: Mechanisms, Efficacy, and Implications

The *w*Mel and *w*AlbB strains of *Wolbachia pipientis*, an intracellular bacterium, inhibit dengue, CHIKUNGUNYA, and Zika virus replication within mosquito cells [235,236,237,238,239]. However, another *Wolbachia* strain, *w*Pip, does not inhibit virus infection in *Ae*. *aegypti* [240] and the mechanism by which *Wolbachia* interferes with virus replication has not been fully elucidated. Current hypotheses include competition between *Wolbachia* and the virus for physical space within mosquito cells and metabolite resources [241,242] and *Wolbachia*-induced modulation of the host’s immune system and immune priming. Immune priming entails sensitizing or preparing the mosquito’s immune system for a faster and more efficient response to a specific pathogen, such as a virus, upon subsequent exposure [243,244].

Despite the lack of a complete understanding of the mechanism or mechanisms involved in *Wolbachia*-associated modulation of viral suppression, the *Wolbachia*-carrying mosquito-based strategy has been deployed as a public health intervention to control dengue transmission (The World Mosquito Program https://www.worldmosquitoprogram.org/) (accessed on 15 February 2023). A randomized study carried out in the city of Yogyakarta, Indonesia, compared the areas where *Ae. aegypti* carrying *Wolbachia* was released with areas without *Wolbachia.* The results revealed a 77% lower incidence of dengue cases, in the *Wolbachia*-treated area [245]. Another study conducted in the city of Niterói, Rio de Janeiro, Brazil, reported a 69% reduction in dengue, 56% in chikungunya, and a 37% reduction in Zika incidence three years after the beginning of the release of *Ae. aegypti* with *Wolbachia* [246].

Although these results bring optimism regarding the use of *Wolbachia* for the control of dengue transmission, these bacteria can have variable effects on mosquito-borne viruses. For example, the *Wolbachia* strain *w*Mel strongly blocked Mayaro virus (MAYV) infections in *Ae. aegypti*, but another strain, *w*AlbB, did not influence MAYV infection in this same vector. *Aedes aegypti* infected with *w*AlbB and *w*Mel showed enhanced Sindbis virus infection rates [247]. The variable effects of *Wolbachia* on vector competence bring into question the safety of the current release of *Wolbachia*-infected mosquitoes. Furthermore, the potential impact of these bacteria on biodiversity has not been thoroughly investigated [248,249], and the risk of the emergence of DENV variants that escape virus-specific inhibition in *Wolbachia*-infected mosquitoes [250,251], underscores the importance of further research on interactions between *Wolbachia*, mosquitoes, viruses, and other organisms.

### 4.3. Symbiotic Microorganisms and Paratransgenesis

Paratransgenesis involves the colonization of vector insects with genetically engineered symbiotic microorganisms that are effective in inhibiting parasite development [252,253,254,255]. Ideal symbionts for effective paratransgenesis are easily manipulated genetically, colonize mosquitoes efficiently, spread into mosquito populations (vertical and horizontal transmission), and are efficient in inhibiting pathogen development in mosquitoes [256]. Proof-of-principle experiments performed primarily with bacteria demonstrated that genetically modified microorganisms expressing antipathogen molecules are capable of interfering with or blocking the development of malaria parasites in mosquitoes [257,258,259]. Among the mosquito symbiotic bacteria, strains of *Asaia*, *Pantoea*, *Serratia*, *Pseudomonas*, and *Thorsellia* have been evaluated as candidates for paratransgenesis [260,261,262,263,264]. 

The fungus *Metarhizium anisopliae* has been genetically transformed to express anti-*Plasmodium* proteins. Mosquitoes treated with transgenic *M. anisopliae* had 71–98% fewer sporozoites present in their salivary glands [204]. Scorpine, one of the molecules expressed by transgenic *M. anisopliae*, also affects negatively dengue virus replication, expanding the application of genetically transformed fungi to control arbovirus transmission [265].

Densovirus, a small DNA virus (4–6 kb) belonging to the Parvoviridae family (Densovirinae subfamily), infects arthropods such as mosquitoes and is maintained in natural populations through both horizontal and vertical transmission from infected adults to larvae. With one of the smallest known viral genomes, Densovirus serves as a valuable molecular tool. Its compact genome can be inserted into an infectious plasmid, which can then be used to express antiparasitic genes, both in cell cultures and in live mosquitoes [266,267,268,269].

The discovery of mosquito symbiotic bacteria [258,270,271,272], viruses [266,267,268,269], and fungi [273] is an active area of research. Advances toward deploying paratransgenesis as a tool for blocking pathogen transmission by mosquitoes also include the identification of antipathogen effector peptides and mechanisms of cellular secretion [253,258]. In bacteria, one efficient way of secreting effector molecules from the bacterial cytoplasm into the lumen of the mosquito intestine was engineered using *Escherichia coli*’s hemolysin-A secretion system. This process involves the export of proteins, such as hemolysin HlyA, through a specific mechanism that includes an export signal at the C-terminal end of the protein, along with the membrane proteins HlyB, HlyD, and the outer membrane protein TolC [274]. The use of bacteria for paratransgenesis is the most advanced alternative compared with viruses and fungi applications. However, concerns about the safety of releasing engineered bacteria into the environment and the potential unforeseen consequences still require attention when contemplating field tests for paratransgenesis. 

Self-limiting paratransgenesis [256] has been suggested as an alternative for initial field trials. This approach proposes the utilization of transient expression of antipathogen compounds from a plasmid that is gradually lost, reverting bacteria to their original wild type. Risk assessment still needs to be carried out and laws and regulations need to be created and enacted before paratransgenesis can be tested in field conditions. However, the processes by which genetically modified microorganisms (GMs) can be spread in nature and how they should act to inhibit the development of target parasites in mosquitoes have already been envisaged. This is illustrated in Figure 3, which presents the paratransgenesis process as a multifaceted approach to combating mosquito-borne diseases.

## 5. Roadmap for the Development of Microbe-Based Products for Controlling Mosquito-Borne Diseases

In this review article, we explored the biotechnological potential of microorganisms for mosquito population control and reduction in vector competence. We listed many microbial agents with mosquito larvicidal activity and provide information on their active metabolites and mechanisms of action. However, most of these mosquitocidal microorganisms and their metabolites have not been developed into new products and marketed as tools and innovations that can be applied to public health interventions. The explanation for the few biolarvicides available on the market is complex and is determined by technical, regulatory, social, and economic factors.

For example, the Organization for Economic Co-operation and Development (OECD) provides information about requirements and approaches to biological pesticides (https://www.oecd.org/env/ehs/pesticides-biocides/data-for-biopesticide-registration.htm) (accessed on 14 July 2023) including a Guidance for Registration Requirements for Microbial Pesticides (https://one.oecd.org/document/env/jm/mono(2003)5/en/pdf) (Aaccessed on 14 July 2023). Accordingly, the Regulation of European Commission (EC) No. 1107/2009 regarding criteria for the approval of microbial pesticides emphasizes the importance of assessing the active substances or the microorganisms themselves for effects on the environment or harmful effects on human or animal health [275]. These directives require collaborative research efforts which may take years to complete. In the United States, the Federal Insecticide, Fungicide, and Rodenticide Act (FIFRA) requires similar assessments, and the U.S. Environmental Protection Agency (EPA) evaluates biopesticides to assure they do not pose unreasonable risks of harm to human health and the environment.

In Brazil [276], the registration of new biopesticides, including biolarvicides, requires evaluation by three federal government agencies that assess them independently and in a specific manner. The Ministry of Agriculture and Livestock (Ministério da Agricultura e Pecuária-MAP, Brasília, DF, Brazil) evaluates efficiency and potential for use in pest control; the Brazilian Institute of the Environment and Renewable Natural Resources (Instituto Brasileiro do Meio Ambiente e dos Recursos Naturais Renováveis-IBAMA, Brasília, DF, Brazil) provides an environmental report; and the Brazilian Health Regulatory Agency (Agência Nacional de Vigilância Sanitária-ANVISA, Brasília, DF, Brazil) conducts the toxicological dossier, assessing the product’s toxicity for the population and the restrictions and requirements for biopesticide use.

The main stages for discovering and developing new larvicides, based on the requirements set forth by government entities, have previously been reviewed [277,278,279]. In summary, the process consists of (1) discovering larvicidal microorganisms; (2) identifying the mechanisms of action of larvicidal microorganisms (live microbial fractions versus metabolites fractions); (3) evaluating human toxicity and pathogenicity of microorganisms and evaluating their effects on non-target organisms and the environment; (4) determining the stability of the candidate larvicide product under field conditions and its shelf life considering its applications in tropical/subtropical, hot and humid environments; (5) comparing the activity of the candidate product with currently available larvicides; and (6) cost analysis (production, storage, transportation, and field application costs) and research of market viability.

Similar considerations will be necessary for the applications designed for reducing disease transmission, such as the deployment of microorganisms with larvicidal and/or antipathogen activity, including paratransgenesis discussed above.

We hope that this review will encourage additional research and investment in the development of new bioinsecticides, highlighting the need to follow the requirements established by regulatory agencies for the approval and registration of products that will assist in the control of mosquitoes and the diseases they transmit.

## 6. Final Considerations

Biotechnological approaches using microorganisms have significant potential to control mosquito populations and reduce their vector competence, making them alternatives to synthetic insecticides. The ongoing research has been crucial in identifying new products and approaches that can be used effectively to control disease transmission. However, the successful implementation of these newly proposed approaches requires a thorough understanding of the multipronged microorganism–mosquito–pathogen–environment interactions. The release of mosquitoes or microorganisms, genetically modified or not, into the environment requires an assessment of the associated risks and benefits. Therefore, the environmental and ethical implications of these proposed releases are active areas of debate [280,281].

Although much has been achieved in discovering new entomopathogenic microorganisms, antipathogen compounds, and their mechanisms of action, reviewed above, only a few have been turned into viable products for mosquito control. There is a discrepancy between the number of microorganisms with potential for the development of new products and the actual available products, highlighting the need for investments in the intersection of research and biotechnology to improve the transition of basic into applied research.

## Figures and Tables

**Figure 1 insects-14-00718-f001:**
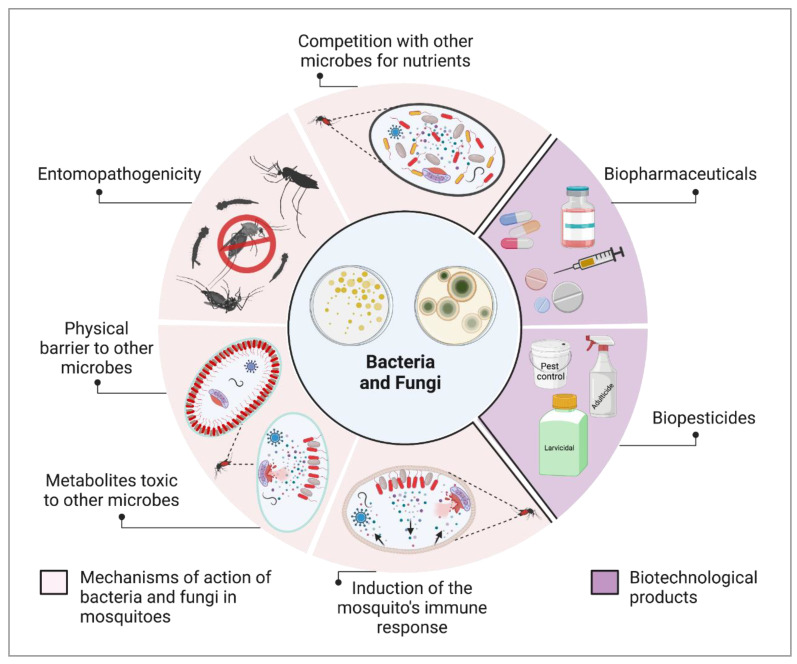
**Microorganisms and their applications for controlling vector populations and disease transmission.** Microorganisms are sources of molecules with insecticide (biopesticides) antipathogen (biopharmaceuticals) activities. Interactions of environmental and symbiotic fungi and bacteria with mosquitoes and their microbiota may affect mosquito and pathogen survival, having implications for vector control and disease transmission. Research that elucidates these interactions is crucial because it underpins the development of novel biotechnological products aimed at effective vector control and reducing disease transmission. Created with BioRender.com (accessed on 5 August 2023).

**Figure 2 insects-14-00718-f002:**
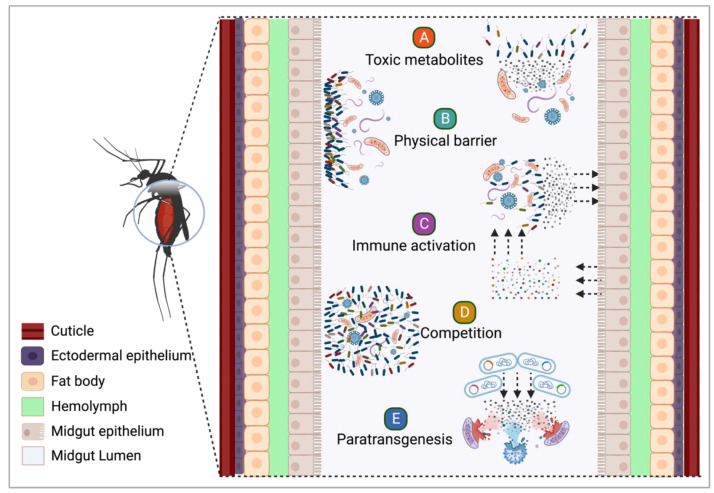
**Biotechnological potential of mosquito symbiotic bacteria against infectious agents**. (**A**), secretion of toxic substances that either kill or arrest the development and replication of viruses and parasites. (**B**), formation of physical barriers through large population accumulation or rearrangements of molecules secreted into the midgut lumen, preventing the passage of parasites to organs essential for their successful development. (**C**), activation of the mosquito immune system, which not only reduces the load of symbiotic bacteria but also leads to the elimination of invading parasites through the secretion of toxic molecules, preventing their propagation in the mosquito’s body. (**D**), competition with infectious agents for space and nutrients can have dire consequences for these pathogens as they must compete with a vastly larger population of symbiotic bacteria in the mosquito’s midgut lumen. This results in limited resources for the pathogens, ultimately leading to their decreased survival and replication within the mosquito. (**E**), paratransgenesis involves populating vector insects with genetically engineered symbiotic microorganisms that effectively hinder the development of parasites through synthesizing and secreting antipathogen molecules. This topic is further explored in topic 4.3 of this review. Created with BioRender.com (accessed on 5 August 2023).

**Figure 3 insects-14-00718-f003:**
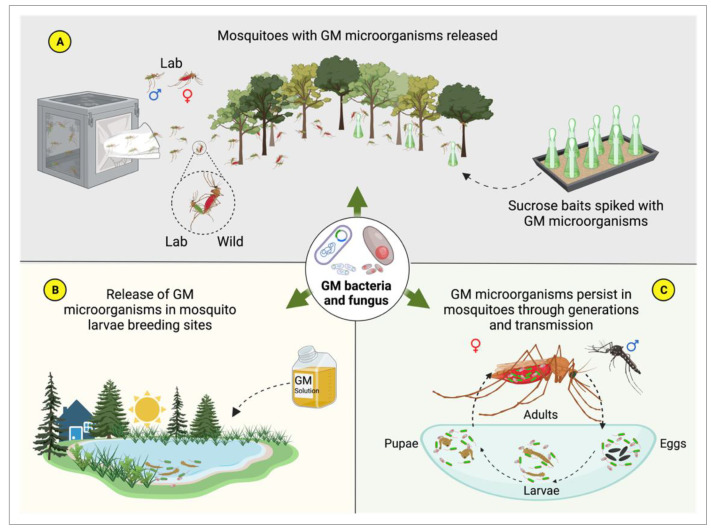
**Strategies for dissemination of GM microorganisms in the wild and their continual circulation among mosquitoes**. (**A**) Male and female mosquitoes are fed in the laboratory with a sucrose solution containing the GM microorganisms and then released into the wild to mate with other wild mosquitoes. The spread of GM microorganisms can also occur through the provision of sucrose baits in the field enriched with GM microorganisms. This enables the GM microorganisms to be transmitted forward, allowing them to spread throughout the wild mosquito population and aiding in reducing vector-borne disease transmission. (**B**) Release of GM microorganisms into natural larval breeding sites. The GM microorganisms are ingested by the larvae and remain associated with them until adulthood. If mosquitoes become infected with a parasite that is a target of the effector molecules produced by the GM microorganisms, these molecules will interfere with the development of the target pathogen, thereby preventing its transmission. (**C**) Persistence of GM microorganisms in mosquitoes for generations through vertical and horizontal transmission. Vertical transmission occurs from parents to offspring, while horizontal transmission takes place between mosquitoes during mating or sharing of breeding sites. The presence of GM microorganisms can continue to impact mosquito populations for an extended period. Created with BioRender.com (accessed on 5 August 2023).

## Data Availability

The data presented in this study are openly available in this manuscript.

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
