# Peer review of "Biotechnological Potential of Microorganisms for Mosquito Population Control and Reduction in Vector Competence"

_insects, 2023, doi:10.3390/insects14090718_

Round 1
Reviewer 1 Report
The manuscript presents a good overview of microorganisms useful in reducing mosquito populations. I suggest the authors better separate paragraph 4. The title defines vector-associated bacteria but paragraph 4.2 describes fungi. It would be helpful to describe Wolbachia in a separate sub-section or put it in 4.3 of symbiotic microorganisms
Author Response
Response to Reviewer 1 Comments
Point 1: The manuscript presents a good overview of microorganisms useful in reducing mosquito populations. I suggest the authors better separate paragraph 4. The title defines vector-associated bacteria but paragraph 4.2 describes fungi. It would be helpful to describe Wolbachia in a separate sub-section or put it in 4.3 of symbiotic microorganisms.
Response 1: The title of section 4 was changed to vector-associated microbiota. We agreed in removing the fungi subtopic and transferring it to section 3, which is now renamed " Fungi and promising approaches for controlling vector mosquitoes and Plasmodium spp.." The Wolbachia subtopic, was maintained under symbiotic microorganisms but given a subtitle, as suggested,"4.2. Wolbachia-based Strategy for Controlling Mosquito-Borne Viruses: Mechanisms, Efficacy, and Implications."
Reviewer 2 Report
Dear authors, your article is very well documented and very interesting. Congratulation! My only recommendation is that the first time a species is mentioned in the text, its name should be accompanied by the name of the author and the year of the description. Ex: Aedes albopictus (Skuse, 1895).
Author Response
Response to Reviewer 2 Comments
Point 1: Dear authors, your article is very well documented and very interesting. Congratulation! My only recommendation is that the first time a species is mentioned in the text, its name should be accompanied by the name of the author and the year of the description. Ex: Aedes albopictus (Skuse, 1895).
Response 1: We appreciate your recommendation, and we have now included the author's name and year of description for all mosquito species mentioned in the review. As a result, the species mentioned for the first time in the article are described as follows: Aedes aegypti (Linnaeus, 1762); Aedes albopictus (Skuse, 1894); Anopheles coluzzii (Coetzee et al., 2013); Anopheles stephensi (Liston, 1901); and Anopheles gambiae (Giles, 1902).
Reviewer 3 Report
Summary
The authors summarised bacterial and fungal microorganisms with potential to be developed into novel biological insecticides, along with their formulations and target mosquito genera. They also gave few and brief examples of published research in using microorganisms to reduce mosquito population or to modulate vector competence and does not go into much detail in terms of mechanisms of action. The review contains an extensive list of relevant references but is lacking in the synthesis of original viewpoints and new insights beyond restating the conclusions of the primary research articles or the reviews that it cites. It needs more work to be able to add value to the field because in its current state, it raises questions rather than answer them. Although well-written with solid grammar, the manuscript could also benefit from rearrangements of certain discussion points for a more cohesive reading experience.
Major comments
The Simple Summary should not be a reiteration of the Abstract. It should instead present the points put forward by the authors in this review.
The Abstract sets up the review to focus only on biological insecticides, but half of the review is on microbial modulation of vector competence.
In Figure 1, the authors list several mechanisms of action, but only two of these (entomopathogenicity and toxic metabolites) are later focused on in this review. It’s not clear whether this is by choice, or if there are no microorganism candidates that belong to the other three mechanisms mentioned. It is also not explained what the authors define as biopharmaceuticals and biopesticides. Adding a title to Figure 1 might help understand what it is about.
Table 1 seems to include formulations that are already developed as registered trademarked products. These seem out of place in a list of potential larvicidal bacteria to explore by further research. It would still be useful to know about these registered larvicidal products but they should be listed separately or have their technology readiness level indicated somehow to distinguish from the the other bacterial species/strains.
There are instances where examples were given of potential microorganism candidates but the authors refrained from going into details, instead summarising the primary research article into one sentence:
"Bacillus mojavensis kills Ae. aegypti larvae and its action was provisionally attributed to the biosurfactant surfactin thioesterase.”;
“The metabolites isolated from M. anisopliae are also active against Ae. aegypti, An. stephensi, and Cx. quinquefasciatus.”;
Lines 317-321;
“The secretion of effector molecules from the cytoplasm of bacteria into the lumen of the mosquito intestine has been engineered using the Escherichia coli hemolysin-A secretion system”
I needed to go to the references themselves to find out how the microorganism achieves this effect. The simplest way of adding value to this manuscript is if the authors explained the discovery in more biological detail.
Line 301-303. “However, other studies did not show an impact of B. bassiana or M. anisopliae on the development of Plasmodium species in Anopheles mosquitoes.” This needs to be expanded. What could account for the differences in the different impacts? How were the experiments different in the studies? What further research could be done to resolve these contradicting conclusions?
Lines 326-327. “The discovery of mosquito symbiotic bacteria [256,264–266], viruses [267–270] and fungi [271] is an active area of research.” This sentence seems out of order here, or is it an underdeveloped point? The 4 references about densovirus paratransgenesis were not discussed at all but they really should be.
Figure 3: It is not clear whether the processes outlines here are from specific cases from the literature or from the authors own considerations. In any case, the figure is under-discussed and leaves some questions open:
Does 3A only apply to GM organisms that are sexually transmitted from males to females? Why not release lab-reared females colonised with the GM organism? Or provide sucrose baits in the field spiked with GM organisms?
Is 3C only applicable to GM organisms producing effector molecules that cause membrane rupture? Can there be other mechanisms of action?
It is also not clear what this figure represents, as these concepts are a mix of release strategies and mechanism of actions.
Some outstanding questions I have at the end of reading this review:
Which of these microorganism/their metabolites are safe for non-target organisms?
Where do most candidates get blocked in the development pipeline, i.e., what are the biggest challenges in biopesticide development?
Are there microorganisms that kill mosquitoes or inhibit pathogens through nutrient competition, forming physical barriers, or inducing the mosquito’s immune response?
Why are biological insecticides preferable over chemical insecticides, when the former can also be dangerous to non-target organisms?
From the abstract, I expected to gain insights on the specific areas further research needs to be directed to, or the main obstacles in developing biological pesticides. Section 5 did not really deliver on these expectations. Indeed, I was left to look at references 277 and 278 for more substantial discussion. Overall, the review should do more to convince why bacterial and fungal microbes are an attractive method of mosquito control.
Minor comments
Consistency in using the terms “pesticides” vs “insecticides” would be desirable.
Table 2 needs to be cited somewhere in Section 3.
General copy-editing is needed, e.g.:
Consistency in capitalising chikungunya virus;
Formatting/text justification in tables;
Out-of-place comma in “Fungi,” (Line 123);
“Yogiakarta” should be Yongyakarta;
“Ae aegypti infected with Wolbachia” (line 255)
Line 90. Table 1 should be cited at the end of the sentence “Despite these limitations, bacterial strains that are suitable for cultivation and have larvicidal activity have been identified.”
Line 272-278. This paragraph looks like it belongs in Section 2 as it describes an effect on mosquito population rather than the infectious agent transmitted by the mosquito. Also, it needs to be made clearer that the two sets of numbers “7.5 million and 14.4 million” and “69% and 95%” are associated with two separate releases in 2017 and 2018. Lastly, is the population suppression rate in 2017 69% or 68% (see first paragraph of Discussion on referenced paper)?
Line 304-310. This paragraph looks like it belongs in Section 4.3 as it is about a paratransgenic fungal candidate modified to express anti-pathogenic effectors.
Reviewer 4 Report
Katak et al. present a review on the use of microorganisms and genetic modification for the control of medically important species of mosquitoes. The topic is within the scope of Insects.
The manuscript requires major improvement before it would be suitable for publication.
Major points.
I) The text is generally superficial and lacks important information in many places. A superabundance of cited references contrasts with a paucity of detail and few insights.
II) Section 2. Describes pathogenicity and toxicity of some bacteria but fails to review their efficacy in mosquito control. The worldwide use of spinosad receives no mention. This is a major oversight.
III) Section 3. Describes use of fungi against mosquitoes but does not address their efficacy in vector control, how widely used they are, or the issues that limit their effectiveness.
IV) The role of formulation in the delivery and efficacy of microorganisms and/or their metabolites is completely overlooked.
V) Regulatory issues appear to be a major limitation but suggestions of how to overcome these are not addressed.
Specific points.
1. Simple summary. Half the text is preamble. Please focus on the scope and findings of the review.
2. Abstract. This is practically a slightly modified cut-and-paste of text from the Simple Summary. Please highlight your principal findings and insights.
3. L73-83. Bt toxins. What about the other toxins like Vip and other soluable proteins? Should you explain the difference between Cry and Cyt? And potential issues with beta-exotoxin producing strains?
4. L108. I did not understand the phrase that begins ....instigates their exploration.... Please reword.
5. Section 2 (Bacteria). I was disappointed to find no mention whatsoever of spinosad (or spinetoram). This is now one of the most important larvicides against Aedes spp. (more than Bti) in many dengue endemic regions.
6. Table 1. Delete "Toxic" in the column on Formulation (line below line 119).
7. L151-153. What types of metabolites are you talking about here? How do they exert toxicity? Are they selective or not?
8. Figure 2. The term "Innate immunity" in the figure should be changed to "Immune activation", as innate immunity is mainly mediate through apoptosis in insects.
9. L238. The concept of paratransgenesis needs to be briefly explained in the Fig 2 legend. The text "If this approach.....vector borne diseases" between lines 239-241 should be deleted.
10. L249. The concept of immune priming needs to be explained briefly.
11. L308-310. I can think of many countries in the world where the release of GMO fungi would be prohibited or highly problematic.
12. L349. Please delete text "This would result......in the region" as this is obvious.
13. L390-392. Delete text "Like in Europe.....efficacy and safety". As this is obvious and already stated at the top of p17.
14. L404-406. Can you provide information on how regulation authorities dealt with the examples given previously, such as Wolbachia? I note that the term "Risk assessment" only appears once in passing in the entire manuscript.
15. L424. So why are there so few bioinsecticides for mosquitoes? Can you provide any insights?
16. L425-428. Are the pesticide regulatory authorities partially responsible for their attitude and approach to biopesticide registration?
17. The references should be formatted following journal guidelines.
Some minor grammatical and syntax errors.
Round 2
Reviewer 3 Report
The authors have satisfactorily addressed my comments.
Reviewer 4 Report
The authors have improved their manuscript. I believe that it is now suitable for publication.
Requires editing.